# The effects of exercise training on autonomic and hemodynamic responses to muscle metaboreflex in people living with HIV/AIDS: A randomized clinical trial protocol

**Gabriel Gama**[1,2], **Marcus Vinicius dos Santos Rangel**[1,2], **Vanessa Cunha de Oliveira Coelho**[1], **Gabriela Andrade Paz**[1,2], **Catarina Vieira Branco de Matos**[1], **Bárbara Pinheiro Silva**[1], **Gabriella de Oliveira Lopes**[1], **Karynne Grutter Lopes**[1,3], **Paulo Farinatti**[1,2], **Juliana Pereira Borges**[1,2]*

1 Laboratory of Physical Activity and Health Promotion, Institute of Physical Education and Sports, University of Rio de Janeiro State, Rio de Janeiro, RJ, Brazil, 2 Graduate Program in Exercise and Sports Sciences, University of Rio de Janeiro State, Rio de Janeiro, RJ, Brazil, 3 Graduate Program in Clinical and Experimental Physiopathology, Faculty of Medical Sciences, University of Rio de Janeiro State, Rio de Janeiro, RJ, Brazil

* julipborges@gmail.com

## Abstract

### Background

People living with HIV (PLHIV) present impaired muscle metaboreflex, which may lead to exercise intolerance and increased cardiovascular risk. The muscle metaboreflex adaptations to exercise training in these patients are unknown. The present study aims to investigate the effects of a supervised multimodal exercise training on hemodynamic and autonomic responses to muscle metaboreflex activation in PLHIV.

### Methods and design

In this randomized clinical trial protocol, 42 PLHIV aged 30–50 years will be randomly assigned at a ratio of 1:1 into an intervention or a control group. The intervention group will perform exercise training (3x/week during 12 weeks) and the control group will remain physically inactive. A reference group composed of 21 HIV-uninfected individuals will be included. Primary outcomes will be blood pressure and heart rate variability indices assessed during resting, mental stress, and activation of muscle metaboreflex by a digital sphygmomanometer and a heart rate monitor; respectively. Mental stress will be induced by the Stroop Color-Word test and muscle metaboreflex will be activated through a post-exercise circulatory arrest (PECA) protocol, being the latter performed without and with the application of a capsaicin-based analgesic balm in the exercised limb. Secondary outcomes will be heart rate, peripheral vascular resistance, stroke volume, cardiac output, blood lactate, anthropometric markers and handgrip maximal voluntary contraction. The intervention and control groups of PLHIV will be evaluated at baseline and after the intervention, while the HIV-uninfected reference group only at baseline.

**Funding:** This project was funded by the Carlos Chagas Filho Foundation for the Research Support in the State of Rio de Janeiro (FAPERJ) in the form of grants to JB [E-26/010.100935/2018, E-26/202.720/2019] and PF [E-26/202.880/2017], and by the National Council for Technological and Scientific Development (CNPq) in the form of a grant to PF [303629/2019-3]. The funders had no role in study design, data collection and analysis, decision to publish, or preparation of the manuscript.

**Competing interests:** The authors have declared that no competing interests exist.

## Discussion

The findings of the present study may help to elucidate the muscle metaboreflex adaptations to exercise training in PLHIV.

## Trial registration

This study will be performed at University of Rio de Janeiro State following registration at ClinicalTrials.gov as NCT04512456 on August 13, 2020.

## Introduction

The Acquired Immunodeficiency syndrome (AIDS) is a chronic, potentially life-threatening condition caused by the Human Immunodeficiency Virus (HIV). Globally, 680,000 people have died from AIDS-related illnesses in 2020 and 1.5 million new infections occurred in that year [1]. Access to combined antiretroviral therapy (cART) has prevented around 12.1 million AIDS-related deaths since 2010, so that 38 million people are currently living with HIV [2].

Despite the increase in life expectancy, the combination of prolonged HIV infection and cART treatment may damage several physiological systems, thereby increasing the risk of non-AIDS related diseases [3, 4]. A recent review [4] revealed that cardiometabolic diseases consistently appeared among the most reported comorbidities in people living with HIV (PLHIV), whose rates of myocardial infarction, heart failure, pulmonary hypertension, sudden cardiac death and stroke are higher in comparison with uninfected subjects [5]. Additionally, increased sympathetic activity, which is a marker of poor cardiovascular prognosis [6] is often found among PLHIV at rest [7, 8] and after physical exercise [9].

During exercise, hemodynamic function mediated by autonomic activity is mainly regulated by the exercise pressor reflex (EPR) [10], which responds to muscle afferent signals from type III (mechanoreflex) and IV (metaboreflex) nerve endings [11, 12]. Those afferent neurons are sensitive to mechanical stimuli and metabolite accumulation, respectively [13]. The EPR increases the sympathetic stimulation to both heart and blood vessels, in order to meet the augmented metabolic demand of working muscles [14]. If muscle mechano- or metaboreflex is blunted, the increase in blood pressure (BP) may not be enough to assure appropriate muscle blood flow and oxygen supply, leading to premature fatigue and exercise intolerance [15, 16]. Considering that abnormal skeletal muscle reflexes play a significant role in limiting the exercise tolerance, as well as in increasing the cardiovascular risk during or after exercise [14], investigations on strategies capable to counteract those impairments are clinically relevant.

Exercise training seems to improve sympathetic activity control and/or abnormal cardiovascular responses to muscle metaboreflex activation in several chronic conditions [17–20]. Although very little information is available on the potential mechanisms by which exercise training favors the muscle metaboreflex, they seem to include a wide spectrum of adaptations, as increased muscle perfusion with lower metabolite accumulation, greater antioxidant capacity, and changes in the expression of receptors evoking metaboreflex (i.e., the transient receptor potential vanilloid receptor 1 –TRPv1) [21]. Individuals with impaired metaboreflex have deficiencies in one or more of those mechanisms, and therefore might be more responsive to exercise training than healthy individuals [21]. This is the case of PLHIV, who often present lactic acidosis [22] and attenuated pressor response to metaboreceptor stimulation [23].

Despite this, we could not find research investigating muscle metaboreflex adaptations and its underlying mechanisms to exercise training in PLHIV. This would be nonetheless useful to determine the potential impact of exercise training on mechanisms controlling autonomic modulation during exercise, which have prognostic value [24]. Thus, the aim of the present study is to describe the experimental protocol of a randomized controlled trial designed to investigate the effects of a multimodal supervised exercise program (i.e., aerobic, resistance and flexibility exercises) on hemodynamic and autonomic responses to muscle metaboreflex activation in PLHIV. In addition, potential mechanisms of exercise training adaptations in the muscle metaboreflex will be analyzed by assessing blood lactate concentration and activating TRPv1. We will test the hypothesis that a multimodal exercise training would be able to increase the neurocardiovascular responses during metaboreflex activation by reducing blood lactate levels at rest and increasing TRPv1 sensitivity.

## Materials and methods

### Study design

This single-center randomized clinical trial will be performed at the Laboratory of Physical Activity and Health Promotion (LABSAU) at the University of Rio de Janeiro State (RJ, Brazil), where a supervised exercise program particularly designed for PLHIV is held since 2005. PLHIV will be recruited in the outpatient clinic of the University hospital, and uninfected volunteers matched for age, sex and body mass will be randomly recruited from the staff of the same institution to compose a reference group. All subjects will receive oral and written instructions about the study risks and benefits and will be included in the study only after giving written informed consent. Recruitment, pre-participation screening and data collection will occur between February 2022 and December 2023. We believe that results will be available for analysis in 2024.

The trial was registered at ClinicalTrials.gov (registration number: NCT04512456, date of registration: August 13, 2020) and approved by the Ethics Committee of the Pedro Ernesto University Hospital (approval number 41955620.2.0000.5259, opinion number 4.520.721, approved in February 2, 2021). The study complies with the SPIRIT 2013 recommendations (Standard Protocol Items: Recommendations for International Trials) (supporting information) [25]. The Fig 1 depicts the overall schedule and time commitment for trial participants [25].

### Participants and recruitment

Eligible PLHIV will be men and women aged 30- to 50 years, diagnosed with HIV/AIDS [26], but asymptomatic and free from opportunist infections. Exclusion criteria will be: a) use of cART for less than 3 years; b) regular physical exercise ($\geq$ 3 days/week during 30 min) in the last 6 months; c) current or past diagnosis of hypertension, coronary artery disease, ischemic disease, pulmonary disease, diabetes mellitus, Chagas disease, tuberculosis, or heart failure; d) malnutrition; e) use of antidepressant or antihypertensive medication, and f) smoking habit. Uninfected controls will be screened for items b, c, d, e, and f.

Forty-two PLHIV will be randomized through a random code generator software (www. randomization.com) at a ratio of 1:1 into intervention or control groups. Therefore, the experiment will have three groups: a) PLHIV under training (PLHIV-T; n = 21), which will enroll in the multimodal exercise training during 12 weeks; b) PLHIV controls (PLHIV-C; n = 21), which will remain in clinical care and shall not to change their physical activity behavior; c) Uninfected controls (CONT; n = 21), which shall not change their physical activity behavior. For ethical reasons, the PLHIV-C group will have the possibility to undertake the exercise

| | STUDY PERIOD | | | | |
|---|---|---|---|---|---|
| | Enrolment | Allocation | Post-allocation | | |
| **TIMEPOINT** | $-t_1$ | $t_0$ | $t_1$ | $t_2$ | $t_3$ |
| **ENROLMENT:** | | | | | |
| **Eligibility screen** | X | | | | |
| **Informed consent** | X | | | | |
| **Allocation** | | X | | | |
| **INTERVENTIONS:** | | | | | |
| *Exercise training (intervention group)* | | | ◄————————► | | |
| *No intervention (usual care)* | | | ◄————————► | | |
| **ASSESSMENTS** | | | | | |
| *Blood pressure* | | X | | | X |
| *Autonomic indices* | | X | | | X |
| *Hemodynamic parameters* | | X | | | X |
| *Blood lactate concentration* | | X | | | X |
| *Anthropometry* | | X | | | X |
| *Handgrip maximal strength* | | X | | | X |

**Fig 1. Schedule of enrolment, interventions, and assessments.** $-t_1$, enrolment week 0; $t_0$, allocation and baseline week; $t_3$, post-intervention (week 12).

training intervention once the trial ends. Outcomes in PLHIV-T and PLHIV-C will be assessed before randomization at baseline and after intervention, whereas in CONT only at baseline to provide reference values. The study flowchart is presented in Fig 2.

## Experimental design

Assessments will be undertaken in a single visit, as shown in Fig 3. All procedures will take place at the same time of the day (9–11 am) to minimize circadian effects, in a quiet temperature-controlled environment (21–23˚C). Participants will be instructed to fast for 2 h, to avoid physical exercises in the 48 h, and caffeine or alcohol in the 12 h prior to experimental sessions.

After a general medical examination and anthropometric measurements, participants will remain 30 min at rest in a supine position before performing a mental stress task and handgrip strength test. The mental stress task will provide information about the integrity of pathways

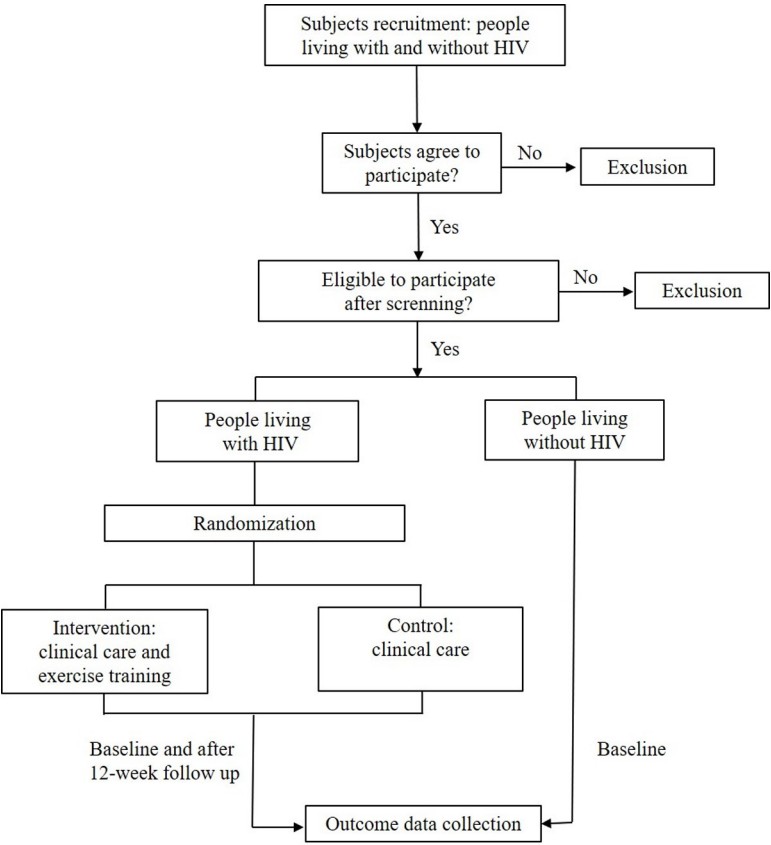

**Fig 2. Study flowchart.**

of stress-induced sympathetic responses mediated by the central command. Maximal voluntary contraction (MVC) assessed by the handgrip test will be used to determine the loads applied during the metaboreflex activation protocol, and to detect possible strength differences between groups at baseline and after intervention. Subsequently, the muscle metaboreflex will be activated using a post-exercise circulatory arrest (PECA) protocol, during which blood

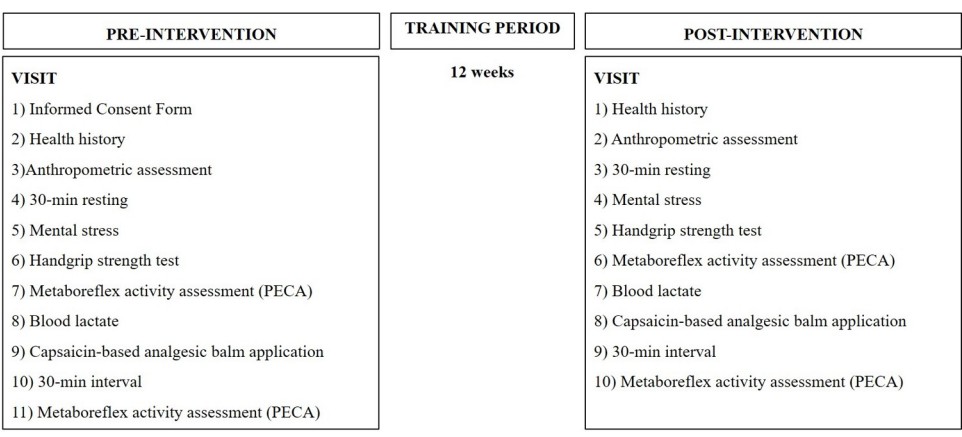

**Fig 3. Experimental design.**

lactate will be assessed. Then, a capsaicin-based analgesic balm (0.1%, FarmaSite, Itaúna, Brazil) will be applied over the exercising arm to stimulate the metabolically sensitive muscle afferents via TRPv1 receptors [27], and after 30 min of topical application, PECA will be repeated.

Primary outcomes will be BP and heart rate variability (HRV) indices reflecting autonomic modulation. Secondary outcomes will be the heart rate (HR), peripheral vascular resistance (PVR), stroke volume (SV), cardiac output (CO), blood lactate concentration, anthropometric markers, and handgrip MVC.

## Intervention

Patients included in the intervention arm will participate of an exercise training program during 12 weeks, consisting of 60-min sessions performed 3 days/week and including aerobic, resistance, and flexibility exercises, as described elsewhere [28]. Treadmill or cycle ergometer aerobic exercise will last 30 min with an intensity range corresponding to 50–80% of the heart rate reserve. Resistance training will consist of 20 min of exercises for the major muscle groups of the upper and lower limbs (8–10 single and multi-joint exercises with free weights and machines), performed with 2–3 sets of 10–12 maximum repetitions (RM) and load corresponding to 80–90% of 10 RM. At the end of the sessions, the participants will perform a 10-min static stretching routine involving major joints (2 sets of 6- to 8 exercises, holding the maximal range of motion for 30-s) [28–30].

Heart rate will be monitored to ensure proper aerobic exercise intensity (Polar H10; Polar Electro$^{TM}$, Kempele, Finland). All training sessions will be performed indoors (21–23˚C), always in the morning and under the supervision of the research staff. The participants will be instructed on the importance of clinical follow-up and the benefits of regular physical activity to improve adherence to intervention.

## Mental stress test

To investigate the integrity of muscle metaboreflex efferent pathways, a mental stress assessment will be performed using the Stroop Color-Word test [31], which has been widely used for stress-induced sympathetic effects mediated by the central command, regardless of the muscle metaboreflex activation [32]. The test will be administered for 5 min by a single researcher that will display slides changing every 2 s, showing color names written in different colors. The volunteers will identify the word colors irrespective of the written information. Changes from rest in autonomic and cardiovascular outcomes will be recorded as outcomes.

## Muscle metaboreflex activity assessment

Muscle metaboreflex will be activated by a PECA protocol, which has been shown to trap muscle metabolites in the exercising limb and to maintain stimulation of group IV muscle afferents without the activity of the central command and mechanoreflex [12]. PECA will consist of 13 min, as follows: 5 min of rest, 2 min of isometric handgrip exercise with the dominant arm with a load corresponding to 40% of MVC, 3 min of vascular occlusion, followed by 3 min of recovery. Vascular occlusion will be applied in the exercised limb by inflating a cuff at 240 mmHg 5 s before the end of exercise, and then deflating it in the 3-min recovery. During PECA, the volunteers will be in a sitting position, and shall avoid performing the Valsalva maneuver and any movement, except for the handgrip. Changes from rest during PECA will be calculated for autonomic and cardiovascular outcomes.

Prior to PECA, a brief familiarization with the handgrip exercise will be allowed. Handgrip force exerted will be placed in front of the subjects to guide the contraction intensity. Vascular occlusion in PECA will be applied using a nylon cuff size 11 x 85 cm connected to a pneumatic

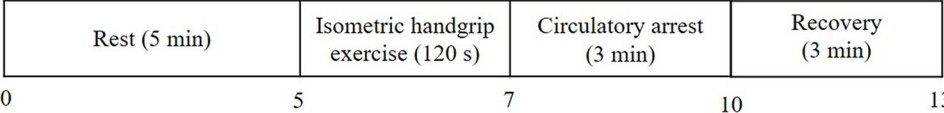

| Rest (5 min) | Isometric handgrip exercise (120 s) | Circulatory arrest (3 min) | Recovery (3 min) |
|:---:|:---:|:---:|:---:|

0                5                7                10                13

**Fig 4. Post-exercise circulatory arrest (PECA) protocol used during muscle metaboreflex activity assessment.**

cuff inflator Hokanson TD312 (D. E. Hokanson[TM] Inc, Bellevue, Washington, USA) placed around the proximal third of the dominant arm. A vascular Doppler probe (DV 610B Medmega Industria de Equipamentos Medicos, Franca, SP, Brazil) will be placed over the brachial artery (dominant arm) to capture the auscultatory pulse and confirm the vascular occlusion during the PECA protocol. The Fig 4 details the PECA protocol.

## Primary outcomes

**Blood pressure.** Systolic and diastolic BP will be assessed in the non-dominant arm using a digital sphygmomanometer (Omron HEM—7113, Kyoto, Japan) positioned around the proximal third of the non-dominant arm. Data will be assessed at rest, and every minute during mental stress and muscle metaboreflex assessments.

**Heart rate variability indices.** The HRV indices reflect cardiac autonomic modulation. Beat-to-beat HR will be continuously recorded using a heart rate monitor (Polar V800; Polar[TM] Electro, Kempele, Finland) [33] at rest, during the mental stress task, and muscle metaboreflex activation protocol. The signals will be transferred to the Polar Precision Performance Software (Polar Electro, Kempele, Finland). After replacing the non-sinus beat by interpolated data derived from adjacent normal RR intervals, times series data will be exported to Kubios[TM] HRV program for analysis (Version 3.4, University of Kuopio, Kuopio, Finland).

Time domain variables will be: mean normal RR intervals (RRi), standard deviation of normal NN intervals (SDNN), square root of the mean squared successive differences from adjacent RRi (rMSSD), and percentage number of pairs of adjacent RRi differing by more than 50 ms from previous RRi (pNN50). SDNN, rMSSD, and pNN50 are acknowledged to reflect the parasympathetic modulation within short periods [34]. Spectral analysis will be performed by the Fast Fourier transform method, with frequency bands corresponding to low-frequency (LF: 0.04–0.15 Hz), high-frequency (HF: 0.15–0.40 Hz), and total power (TP; meaning LF plus HF) [35]. The LF and HF reflect a predominance of combined sympathetic-parasympathetic and isolated parasympathetic modulation, respectively [35], while LF: HF ratio will be adopted as a marker of sympathovagal balance.

## Secondary outcomes

**Hemodynamic parameters.** BP and HR will be continuously and non-invasively measured on a beat-to-beat basis using finger photoplethysmography (Finometer PRO, Finapres, Enschede, Netherlands). For this, an appropriate size cuff will be placed around the middle finger of the non-dominant hand, which will be supported on a table and positioned at the heart level. BP waveform measured at middle finger will be calibrated for each participant using an automatic cuff placed around ipsilateral arm [36]. SV will be estimated from finger pressure waveform using Modelflow method (BeatScope 2.1, Finometer PRO, Finapres, Enschede, Netherlands) [37]. CO will be obtained multiplying HR by SV, and PVR will be the ratio of mean BP to CO. Brachial BP measured with the digital sphygmomanometer will be used to validate Finometer measurements of absolute blood pressure.

**Blood lactate concentration.** Blood lactate will be measured to verify the metabolic stress induced by PECA. With this purpose, the fingertip of the exercising arm will be punctured with an automatic lancet and 25 µl of capillary blood sample will be collected during PECA, at minutes 5, 7 and 10. The blood sample will be transferred to a microtube containing 50 µl of 1% sodium fluoride, and blood lactate concentration will be determined by the YSL 2700 analyzer (Yellow Spring™ Co., Yellow Springs, Ohio, USA).

**Anthropometry.** Height and body mass will be assessed by a digital scale (Filizola™, Sao Paulo, SP, Brazil), and wall-mounted stadiometer (Sanny™, Sao Paulo, SP, Brazil), respectively. The BMI will be calculated ($Kg/m^2$).

**Handgrip maximal strength test.** Handgrip MVC will be assessed using a hydraulic handgrip dynamometer (Saehan, SH5001, Holliston, MA, USA). Strength measurements will be performed with participants in a sitting position, arm along the body, and elbow in flexion (dominant arm). The volunteers will make three attempts of 5 s interspersed with 2-min intervals, and the highest value will be recorded as result.

## Sample size calculation and statistical analysis

Sample size was calculated using GPower 3.1.9.4 (Universität Kiel, Kiel, Germany) to detect longitudinal differences between groups in systolic BP response due to metaboreflex gain. Considering a between-group difference of 3.5 mmHg (standard deviation of 4.6 mmHg) [23], 80% power and 5% significance level, and increasing the sample size by 50% to account for losses to follow up, a total of 21 participants in each group was calculated as necessary.

Descriptive analysis will consist of mean and standard deviation for continuous variables and percentage for categorical variables. Data normality will be checked by the Shapiro Wilk test, and data will be presented as mean and standard deviation or percentage, whenever appropriate. A cross-sectional comparison between PLHIV-T, PLHIV-C and CONT will be performed at baseline by means of one-way ANOVA (continuous) or chi-squared test (categorical).

Longitudinal effects of exercise training on primary and secondary outcomes will be evaluated through linear mixed models (LMM) including treatment (PLHIV-T or PLHIV-C) and time as fixed effects, and group *vs*. time interaction. LMM, which correlates with repeated measures over the time, allows an intention-to-treat analysis as it includes all observations of each one of the participants, regardless of losses to follow-up or noncompliance to exercise protocol. Residual plots of all models will be examined and the likelihood-ratio test will be used to compare and select random intercept or random slope models. Data analysis will be performed using Stata 13.0 (StataCorp, College Station, TX, USA), and in all cases the statistical significance will be set at $P \leq 0.05$.

## Potential harms

In the event of any clinical intercurrence during the training sessions or experimental protocol, the participants will be referred to the University Hospital for medical care. Any adverse clinical event will be reported to the ethics committee.

## Data management, accessibility, and dissemination policy

A trained research assistant will be responsible for including all primary and secondary outcomes in an Excel spreadsheet in the proper format for analysis, which will be checked by the principal investigator. Unnamed data collected and research results will be deposited in the cloud (Dropbox) and may be consulted after permission granted by the principal investigator. All personal data will be properly stored in folders that will be kept confidential, and will not

be passed on to any third party without prior consent from the participants. At the study end, each participant will receive a full report with the results of their assessments. The results of the study will be publicized in scientific congresses and journals.

## Masking and blinding

The behavioral feature of intervention (exercise training) precludes the blinding of the patients. On the other hand, data analysis will be performed by a single trained researcher, blinded for the group allocation and use or not of capsaicin-based analgesic balm in data collected during PECA.

## Discussion

The findings of the present study may help to elucidate the role of exercise training to produce adaptations in autonomic and hemodynamic responses induced by muscle metaboreflex in PLHIV. Prior research has demonstrated that PLHIV present abnormal pressor response during muscle metaboreceptor stimulation [23] and autonomic dysfunction at rest [38] and after exercise [9]. When group IV muscle afferent feedback is impaired, the increase in BP may not be enough to assure an appropriate muscle blood flow and oxygen supply, leading to premature fatigue and exercise intolerance [16], as previously reported in PLHIV [39].

Most studies addressing the effects of exercise training in PLHIV have focused on physical fitness, body composition, immune function, mental health or lipid profile [40–42], whereas the neurocardiovascular function of these patients has gained little attention. To the best of our knowledge, there are only two clinical trials investigating the role of exercise training on the autonomic function in PLHIV [43, 44]. Quiles et al. [43] have recently demonstrated that an 8-week aerobic exercise program improved the overall autonomic function of these patients, while Pedro et al. [44] did not observe changes in HRV after 16 weeks of concurrent training. Additionally, we could not locate studies investigating the role of exercise training on the muscle metaboreflex among these patients.

In the present protocol, the muscle metaboreflex will be activated by PECA, which is a non-invasive and non-pharmacological method widely used to isolate the metabolic component of ergoreflex, by arresting blood flow of the exercising muscles with a suprasystolic level cuff occlusion that traps the metabolic by-products of exercise within the muscles after the cessation of exercise [45–47]. Of note, PECA will be originally employed in combination to the topical application of a capsaicin-based analgesic balm over the exercising arm. The capsaicin is an active component of chili peppers targeting the TRPv1, which are important mediators of the metabolic stimulation of EPR [27]. Evidence has recently shown that a topical application of a capsaicin-based analgesic balm prior to the test attenuated the pressor and muscle sympathetic nervous activity responses to PECA [27]. However, capsaicin-based analgesic balms to assess exercise training-related adaptations on muscle metaboreflex have never been used. Another important aspect of this study is the multimodal characteristic of the exercise training protocol (e.g., aerobic and resistance exercises), as most studies using this modality of training have found benefits in EPR among other populations [17–19, 48], while those including only resistance training failed to observed similar adaptations [49, 50].

The present study protocol has some limitations. Firstly, autonomic modulation will be assessed by HRV. Although HRV has the advantage of being a simple, non-invasive method capable of assessing dynamic changes in the autonomic control of heart rate [51], more objective measurements of muscle sympathetic outflow, such as microneurography would provide more specific and direct information. Moreover, it will not be possible to control the diet and sleep habits of participants. Despite this, the present study might extend the current knowledge

by demonstrating whether a supervised multimodal exercise training would be capable to improve the muscle metaboreflex among PLHIV. In addition, we also intend to provide new insights into the mechanisms potentially underlying the exercise-induced changes in the muscle metaboreflex, by assessing the influence of TRPv1 receptors, blood lactate concentration, and hemodynamic outcomes (e.g., HR, SV, CO, and PVR).

## Supporting information

**S1 Checklist. SPIRIT checklist.**
(DOC)

**S1 File. Study protocol approved by the ethics committee in English.**
(PDF)

**S2 File. Study protocol approved by the ethics committee in the original language.**
(PDF)

## Author Contributions

**Conceptualization:** Gabriel Gama, Karynne Grutter Lopes, Paulo Farinatti, Juliana Pereira Borges.

**Data curation:** Gabriel Gama, Marcus Vinicius dos Santos Rangel, Vanessa Cunha de Oliveira Coelho, Gabriela Andrade Paz, Catarina Vieira Branco de Matos, Bárbara Pinheiro Silva, Gabriella de Oliveira Lopes.

**Formal analysis:** Karynne Grutter Lopes, Paulo Farinatti, Juliana Pereira Borges.

**Funding acquisition:** Paulo Farinatti, Juliana Pereira Borges.

**Investigation:** Karynne Grutter Lopes.

**Project administration:** Juliana Pereira Borges.

**Supervision:** Juliana Pereira Borges.

**Writing – original draft:** Gabriel Gama, Marcus Vinicius dos Santos Rangel, Gabriela Andrade Paz, Catarina Vieira Branco de Matos, Bárbara Pinheiro Silva, Karynne Grutter Lopes, Paulo Farinatti, Juliana Pereira Borges.

**Writing – review & editing:** Gabriel Gama, Marcus Vinicius dos Santos Rangel, Vanessa Cunha de Oliveira Coelho, Gabriela Andrade Paz, Catarina Vieira Branco de Matos, Bárbara Pinheiro Silva, Gabriella de Oliveira Lopes, Karynne Grutter Lopes, Paulo Farinatti, Juliana Pereira Borges.

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
