## [Decision Letter · Decision Letter 0]

21 Jan 2022

PONE-D-21-31063The effects of exercise training on autonomic and hemodynamic responses to muscle metaboreflex in people living with HIV/AIDS: a randomized clinical trial protocol .PLOS ONE

Dear Dr. Pereira Borges,

Thank you for submitting your manuscript to PLOS ONE. After careful consideration, we feel that it has merit but does not fully meet PLOS ONE’s publication criteria as it currently stands. Therefore, we invite you to submit a revised version of the manuscript that addresses the points raised during the review process.

We look forward to receiving your revised manuscript.

Kind regards,

Walid Kamal Abdelbasset, Ph.D.

Academic Editor

PLOS ONE

Journal Requirements:

2. Thank you for stating the following in the Acknowledgments/ Funding Section of your manuscript: 

The project will be funded by the Carlos Chagas Filho Foundation for the Research Support in the State of Rio de Janeiro (FAPERJ), grants E-26/010.100935/2018 and E-26/202.720/2019 recipient JB) (supplemental file S3). Avenida Erasmo Braga, 118 – 6o andar – Castelo – Rio de Janeiro – CEP 20 020-000. Tel (+55) 21 2333-2000. Fax (+55) 21 2332-6611. e-mail: central.atendimento@faperj.br. Website http://www.faperj.br. The funders had no role in study design, data collection and analysis, decision to publish, or preparation of the manuscript.

The project will be funded by the Carlos Chagas Filho Foundation for the Research Support in the State of Rio de Janeiro (FAPERJ), grants E-26/010.100935/2018 and E-26/202.720/2019 recipient JB) (supplemental file S3). Avenida Erasmo Braga, 118 – 6o andar – Castelo – Rio de Janeiro – CEP 20 020-000. Tel (+55) 21 2333-2000. Fax (+55) 21 2332-6611. e-mail: central.atendimento@faperj.br. Website http://www.faperj.br. The funders will not have a role in study design, data collection and analysis, decision to publish, or preparation of the manuscript.

- https://clinicaltrials.gov/ct2/show/NCT04512456

The text that needs to be addressed involves the Abstract.

In your revision ensure you cite all your sources (including your own works), and quote or rephrase any duplicated text outside the methods section. Further consideration is dependent on these concerns being addressed.

Reviewers' comments:

Reviewer's Responses to Questions

**Comments to the Author**

1. Does the manuscript provide a valid rationale for the proposed study, with clearly identified and justified research questions?

Reviewer #1: Partly

Reviewer #2: Yes

2. Is the protocol technically sound and planned in a manner that will lead to a meaningful outcome and allow testing the stated hypotheses?

Reviewer #1: Partly

Reviewer #2: Yes

3. Is the methodology feasible and described in sufficient detail to allow the work to be replicable?

Reviewer #1: Yes

Reviewer #2: Yes

4. Have the authors described where all data underlying the findings will be made available when the study is complete?

Reviewer #1: Yes

Reviewer #2: Yes

5. Is the manuscript presented in an intelligible fashion and written in standard English?

Reviewer #1: Yes

Reviewer #2: Yes

6. Review Comments to the Author

You may also provide optional suggestions and comments to authors that they might find helpful in planning their study.

Reviewer #1: 1. Short tile is not clear and it looks like key words. Try to make clear.

2. For line 48 and 49 keep the reference.

3. Why the primary outcome is BP and HR. As these will definitely shows changes after the exercise. Can you take other primary outcome measures if possible.

4. in the treadmill will the speed for the all the study participants will be similar or different? What will be the average limitation of speed?

5. sample size looks to be small even including 20% drop out.

6. Why not repeated Measures of ANOVA?

Reviewer #2: If the required modifications done this will be a good study and the results can be generalized, hope you do the recommended modifications .

7. PLOS authors have the option to publish the peer review history of their article (what does this mean?). If published, this will include your full peer review and any attached files.

Reviewer #1: No

Reviewer #2: **Yes: **Marwa Eid

---

## [Author Response · Author response to Decision Letter 0]

7 Feb 2022

Reviewer #1: 

1) Short tile is not clear and it looks like key words. Try to make clear.

Answer: Short title has been changed to “The effect of exercise training on exercise pressor reflex in HIV infection”.

2) For line 48 and 49 keep the reference.

Answer: A reference has been included.

3) Why the primary outcome is BP and HR. As these will definitely shows changes after the exercise. Can you take other primary outcome measures if possible.

Answer: Thank you for your comment. We noticed that the primary outcomes were misreported in the clinical trial record. Instead of blood pressure and heart rate, it should be written blood pressure and heart rate variability. The clinical trial record has been corrected. 

The primary outcomes will be blood pressure (BP) and heart rate variability (HRV), which provide autonomic indices that reflect cardiac autonomic modulation. As people living with HIV often present autonomic dysfunction and impaired cardiovascular responses to metaboreflex activation, it is still unclear whether exercise training will benefit these responses to metaboreflex activation. 

4) In the treadmill will the speed for the all the study participants will be similar or different? What will be the average limitation of speed?

Answer: The treadmill speed will be prescribed individually to achieve the same moderate exercise intensity for all participants (i.e., 50 to 80% of the heart rate reserve), as described in prior work from our group (DOI 10.1080/25787489.2021.1979727, 10.1111/sms.13312, and 10.3389/fphys.2018.01641). Maximal speed will correspond to 80% of the heart rate reserve, which depends on several factors, especially maximal exercise capacity. 

5) Sample size looks to be small even including 20% drop out.

Answer: Thank you for the comment. Sample size has been recalculated, and it is now correct. At the end of the study, a new calculation will be carried out to verify if the necessary power has been achieved.

6) Why not repeated Measures of ANOVA?

Answer: Both repeated measures ANOVA and linear mixed models (LMM) could be used in this study design. However, ANOVA is not as flexible as LMM in handling random missing data, because it only allows listwise deletion, which precludes the intention-to-treat (ITT) analysis and causes bias. The ITT analysis is considered the standard approach to a randomized controlled trial analyses because it typically gives an unbiased estimate of the effect of treatment assignment and reflects the reality that nonadherence occurs in real-world practice. To better address this issue, more information on the LMM analysis has been provided. 

Reviewer #2

If the required modifications done this will be a good study and the results can be generalized, hope you do the recommended modifications.

Answer: Thank you for all your thoughtful comments. All suggestions have been incorporated into the revised document.

1) Line 28 Mention the number of patients.

Answer: The change has been made.

2) Line 31 In your clinical trial registeration the primary and secondary outcome not the same as in the protocol it should be corrected.

Answer: Thank you for this comment. This issue has been corrected.

3) Line 87 Whats ment by multi modal exercise? you shoud add aparagraph to explain it.

Answer: The multimodal supervised exercise program included aerobic, resistance and flexibility exercises. This information has been included in line 88. In addition, more detailed information on the exercise training has been provided in the Intervention section.

4) Line 118 This age range not the same age registered on clincal trial .gov. data should be identical in both protocols.

Answer: Thank you for this comment. This issue has been corrected.

5) Line 125 What is your sample size please mention the sample size and the way of calculation ,as you mention in clinical trial registeration 66 particiants???

Answer: Thank you for this comment. This issue has been corrected. 

6) Line 154 Your out come measures should be the same order in the abstract, method section and clinical trial protocol.

Answer: Thank you for this comment. This issue has been corrected. 

We have noticed that the methods for measuring hemodynamics parameters described in the manuscript were discrepant from the clinical trial record, therefore this issue has also been corrected (Line 231). 

7) Line 169 Add reference for exercise program.

Answer: We have added references for the exercise program.

8) References These references 5,6,,10,12,13,16,17,30,34,37,51 seems to be old you can replace it by more recent reference.

Answer: Thank you for this comment. The referred references have been replaced. 

Journal Requirements

Answer: We have checked the whole manuscript to assure that it meets PLOS ONE's style requirements.

2) Thank you for stating the following in the Acknowledgments/ Funding Section of your manuscript: 

The project will be funded by the Carlos Chagas Filho Foundation for the Research Support in the State of Rio de Janeiro (FAPERJ), grants E-26/010.100935/2018 and E-26/202.720/2019 recipient JB) (supplemental file S3). Avenida Erasmo Braga, 118 – 6o andar – Castelo – Rio de Janeiro – CEP 20 020-000. Tel (+55) 21 2333-2000. Fax (+55) 21 2332-6611. e-mail: central.atendimento@faperj.br. Website http://www.faperj.br. The funders had no role in study design, data collection and analysis, decision to publish, or preparation of the manuscript.

The project will be funded by the Carlos Chagas Filho Foundation for the Research Support in the State of Rio de Janeiro (FAPERJ), grants E-26/010.100935/2018 and E-26/202.720/2019 recipient JB) (supplemental file S3). Avenida Erasmo Braga, 118 – 6o andar – Castelo – Rio de Janeiro – CEP 20 020-000. Tel (+55) 21 2333-2000. Fax (+55) 21 2332-6611. e-mail: central.atendimento@faperj.br. Website http://www.faperj.br. The funders will not have a role in study design, data collection and analysis, decision to publish, or preparation of the manuscript.

Answer: Any funding-related text was removed from the manuscript, and the amended funding statement was included in the cover letter.

3) We noticed you have some minor occurrence of overlapping text with the following previous publication(), which nedds to be addressed:

- https://clinicaltrials.gov/ct2/show/NCT04512456

The text that needs to be addressed involves the Abstract.

In your revision ensure you cite all your sources (including your own works), and quote or rephrase any duplicated text outside the methods section. Further consideration is dependent on these concerns being addressed.

Answer: The abstract and the clinical trial record have been modified to avoid overlapping text.

---

## [Decision Letter · Decision Letter 1]

3 Mar 2022

The effects of exercise training on autonomic and hemodynamic responses to muscle metaboreflex in people living with HIV/AIDS: a randomized clinical trial protocol .

PONE-D-21-31063R1

Dear Dr. Pereira Borges,

We’re pleased to inform you that your manuscript has been judged scientifically suitable for publication and will be formally accepted for publication once it meets all outstanding technical requirements.

Kind regards,

Walid Kamal Abdelbasset, Ph.D.

Academic Editor

PLOS ONE

Additional Editor Comments (optional):

Reviewers' comments:

Reviewer's Responses to Questions

**Comments to the Author**

1. Does the manuscript provide a valid rationale for the proposed study, with clearly identified and justified research questions?

Reviewer #1: Yes

Reviewer #2: Yes

2. Is the protocol technically sound and planned in a manner that will lead to a meaningful outcome and allow testing the stated hypotheses?

Reviewer #1: Yes

Reviewer #2: Yes

3. Is the methodology feasible and described in sufficient detail to allow the work to be replicable?

Reviewer #1: Yes

Reviewer #2: Yes

4. Have the authors described where all data underlying the findings will be made available when the study is complete?

Reviewer #1: Yes

Reviewer #2: Yes

5. Is the manuscript presented in an intelligible fashion and written in standard English?

Reviewer #1: Yes

Reviewer #2: Yes

6. Review Comments to the Author

You may also provide optional suggestions and comments to authors that they might find helpful in planning their study.

Reviewer #1: Thank you for addressing the previous comments. Your primary outcome is heart rate variability and you have mention everything clearly.

Reviewer #2: all the required comments and modification s that had been required had been corrected and added ,it seems that this protocol will be sound for publication

7. PLOS authors have the option to publish the peer review history of their article (what does this mean?). If published, this will include your full peer review and any attached files.

Reviewer #1: No

Reviewer #2: **Yes: **Marwa M.Eid

---

## [Editor Report · Acceptance letter]

10 Mar 2022

PONE-D-21-31063R1 

The effects of exercise training on autonomic and hemodynamic responses to muscle metaboreflex in people living with HIV/AIDS: a randomized clinical trial protocol. 

Dear Dr. Borges:

I'm pleased to inform you that your manuscript has been deemed suitable for publication in PLOS ONE. Congratulations! Your manuscript is now with our production department. 

Kind regards, 

on behalf of

Dr. Walid Kamal Abdelbasset 

Academic Editor

PLOS ONE